# COVID-19 Associated Contact Restrictions in Germany: Marked Decline in Children’s Outpatient Visits for Infectious Diseases without Increasing Visits for Mental Health Disorders

**DOI:** 10.3390/children8090728

**Published:** 2021-08-25

**Authors:** Mara Barschkett, Berthold Koletzko, C. Katharina Spiess

**Affiliations:** 1Department of Education and Family, German Institute for Economic Research (DIW), 10117 Berlin, Germany; mbarschkett@diw.de (M.B.); kspiess@diw.de (C.K.S.); 2Freie Universität Berlin, 14195 Berlin, Germany; 3Department Pediatrics, Dr. von Hauner Children’s Hospital, LMU University Hospital, LMU—Ludwig-Maximilians—Universität Munich, Lindwurmstr. 4, 80337 Munich, Germany

**Keywords:** COVID-19, infectious disease transmission, mental health, school closures, day care center closures

## Abstract

Children have a low risk for severe COVID-19 infections, but indirect consequences of the pandemic may affect their health. We evaluated nationwide data on children’s outpatient visits before and during the first wave of the COVID-19 pandemic in Germany. Data from the National Association of Statutory Health Insurance Physicians for all children with statutory health insurance and at least one physician’s office visit between January 2019 and June 2020 were evaluated for total visits and selected diagnoses for the 2nd quarter of 2019 (8.29 million children, controls) and the 2nd quarter of 2020 (8.5 million, pandemic). Outpatient visits per child fell by 18% during the first wave of the pandemic. Outpatient visits associated with diagnosed infections fell markedly by 51%, particularly for children up to age 5 years for gastroenteritis (73%), otitis media (71%), and streptococcal angina (78%). Outpatient visits for diagnosed chronic physical disorders (diabetes, celiac disease, and hay fever) and mental and behavioral disorders showed little change. Reduced contact between children appears to markedly reduce infection transmission. Infection risks in educational settings should be attenuated after the pandemic through targeted education and counseling and appropriate relationship prevention measures to improve quality of life and opportunities for children and to reduce stress and lost work time for parents.

## 1. Introduction

Children are much less likely to be affected by severe COVID-19 than adults [1]. However, contact restrictions implemented as a consequence of the pandemic and further changes in children’s living conditions may affect their mental and physical health and might lead, for example, to reduced physical activity, unfavorable diets, and increased risk of obesity [2,3]. Markedly altered life conditions with closed day care centers, educational, and recreational facilities and hence markedly reduced peer to peer contact, and a daily routine predominantly restricted to the immediate family environment could alter the risks for infections, mental disorders, or accidents as well as subsequent medical visits. In adults, infectious diseases declined due to contact being limited by the COVID-19 pandemic [4], whereas there is a lack of representative data on children. A decrease in inpatient treatment cases in children by up to 45% has been reported in the United States [5] and by 13–16% in Europe [6]. A 45% decrease in outpatient pediatric treatment cases has been reported in the city of New York [7]. We analyzed nationally collected data on children’s outpatient visits in Germany with respect to the frequency of medical presentations and selected diagnoses before and during the COVID-19 pandemic to evaluate the health impact of the pandemic on children.

## 2. Materials and Methods

We evaluated the nationwide data of the *Kassenärztliche Bundesvereinigung* (KBV) in anonymized form, in compliance with data protection regulations, for the population of all children with statutory health insurance in Germany with at least one visit to a medical doctor’s office between January 2019 and June 2020. In 2019 and 2020, about 90% of all children between 0–15 years were insured via one of the statutory health insurance funds [8,9]. Thus, our data cover most children living in Germany. The analyzed dataset includes the number of treatment cases and, at the patient level, the diagnoses documented with ICD-10 codes [10] as well as the year and month of birth for the birth cohorts for the period of 2007–2019. Note, telemedicine consultations by phone or videoconference, which happened more frequently during the pandemic, are also documented in this database. The frequencies of outpatient treatment cases and selected diagnoses were evaluated for the second quarter (Q2) of 2019 (control period) for children born in the period of 2007–2018 and for Q2 of 2020 (pandemic period) for children born in the period of 2008–2019. We report the average shares of children affected from selected diagnoses for both the control and the pandemic group. Children were categorized into four age groups: “infants” (1–2 years), “preschoolers” (3–5 years), “elementary school children” (6–10 years), and “early adolescence” (11–12 years). Commonly coded physical, mental, and chronic conditions were evaluated. Infectious and parasitic diseases (A00–B99), diseases of the ear and mastoid process (H60–H95), and diseases of the respiratory system (J00–J99) were selected from more than 13,000 different ICD codes [10]. Within these chapters, particularly common or serious subgroups were considered separately: infectious intestinal diseases (A00–A09), diseases of the middle ear and mastoid process (H65–H75), acute upper respiratory tract infections (J00–J06), and streptococcal angina (J03, B95). Furthermore, the chapters for injuries, poisonings, and certain other consequences of external causes (S00–T98) as well as the chapter for mental and behavioral disorders (F00–F99) with the subgroups for neurotic, stress, and somatoform disorders (F40–F48); personality and behavioral disorders (F60–F69); developmental disorders (F80–F89); and behavioral and emotional disorders with onset in childhood and adolescence (F90–F98) were analyzed. Examples of chronic diseases that were evaluated were diabetes mellitus (E10–E14), celiac disease (K90), and hay fever (J30). All outcome variables were binary coded (1 if the diagnosis was made at least once per calendar quarter, 0 if no diagnosis). The percentage change of the respective diagnoses in Q2 2020 compared to Q2 2019 was calculated by comparing means. Statistical significances of differences and 95% confidence intervals were calculated by estimating a binary OLS model (Ordinary Least Square), where a dummy variable that indicates whether the observation was in 2020 (pandemic group) or in 2019 (control group) was regressed on the respective health outcome variables. In addition, age, gender, month of birth, and county fixed effects were included as control variables. We used R software (R Project for Statistical Computing, R Foundation Vienna, r-project.org accessed on 10 July 2021) and chose a *p* value < 0.01 as the statistical significance level.

## 3. Results

Data from 8.29 million children in 2019 and 8.5 million children in 2020 were included. The number of outpatient treatment cases per child fell from 0.71 ± 0.50 (MW ± SD) in Q2 2019 to 0.59 ± 0.52 in Q2 2020 (*p* < 0.0001, 82% of the control period), with a similar decline in all age groups (Table 1). The number of diagnosed infections fell to 49% from the control period (Figure 1 displays the average prevalences in % on the y-axis and the different diseases in the control and pandemic period on the x-axis; exact means, SD, *p*-values, and confidence intervals can be found in Appendix A), with a greater decline in children aged 5 years and younger (to 46% and 41% in 1–2- and 3–5-year-olds, respectively) than in older children (55% and 57% in 6–10- and 11–12-year-olds, respectively) (Table 1). Among outpatient visits for infectious disorders, there was a particularly marked reduction in gastroenteritis, which was down to 27% of the control period (Figure 1), with the greatest reduction in 3–5-year-olds (23%), otitis media with other middle ear and mastoid diseases in children aged 1–2 and 3–5 years (to 22% and 28%, respectively), and streptococcal angina reduced to 22%, with comparable reductions in all age groups except for a slightly smaller decrease in 1–2-year-olds (Table 1).

The frequency of outpatient visits associated with diagnosed injuries fell to 83% in the overall group (Figure 1). The decrease primarily occurred in school-aged children (6–10 years: 80%, 11–12 years: 67%), while there was little change in preschool children (1–2 years: 90%, 3–5 years: 97%) (Table 1).

In contrast to infections, the number of outpatient visits resulting in diagnoses of common chronic diseases decreased to a lesser degree than the total number of outpatient visits for diabetes (to 92%), celiac disease (to 86%), and hay fever (to 95%) (Figure 2), although the changes in the age group for the children aged 1 to 5 years were often not statistically significant (Table 1). A greater decrease in hay fever diagnoses to 70% of the comparison period was observed in infants (Table 1), in whom, however, differential diagnosis between allergic rhinitis and persistent infectious rhinitis may be more difficult.

The number of outpatient visits associated with diagnosed mental and behavioral disorders dropped to 89% of the control period (Figure 3), with a trend toward a lesser reduction in younger children aged 1–2 years (94%) and 3–5 years (91%) (Table 1). The frequency of subgroup diagnoses captured by ICD coding (Neurotic, Stress, and Somatoform Disorders; Personality and Behavioral Disorders; Developmental Disorders; Behavioral and Emotional Disorders with Onset in Childhood and Adolescence), was proportional to the total diagnoses of mental and behavioral disorders.

## 4. Discussion

The data of more than 8 million children aged 1 to 12 years with statutory health insurance evaluated here correspond to approximately 90% of children in this age group in Germany [11]. We evaluated the data for the period from April to June 2020 compared to the corresponding period of the previous year to determine what changes occurred in regard to outpatient visits and diagnosed disorders among children during and after the first COVID-19 related contact restrictions. Thus, we considered the period of the COVID-19 pandemic when contact restrictions were introduced in Germany on 22 March 2020, leading to the widespread closure of schools and day care centers until early May 2020.

In the observation period during the first wave of the COVID-19 pandemic in Germany, the number of outpatient visits to medical doctors’ offices decreased by almost one fifth. This decrease may be due to the avoidance of non-urgent doctor visits because of parental concerns about risks of infection in a doctor’s offices but may also be due to a reduced burden of disease associated with contact restrictions. The observed decrease in the number of cases is significantly lower than the 45% decrease reported from New York City [7], suggesting a higher level of family confidence in the safety precautions taken in physicians’ offices in Germany. This is also supported by the fact that the number of presentations for chronic physical illnesses, the prevalence of which is unlikely to have been influenced by the pandemic in the short term, decreased only slightly (diabetes by 8%, celiac disease 14%, hay fever 5%). Apparently, families tended to visit their doctor if there were serious disease burdens despite the pandemic associated risks.

In contrast to these chronic physical disorders, there was a huge decline in outpatient visits for diagnosed infectious diseases to less than half the frequency in the pre-pandemic period, with a particularly pronounced reduction in the diagnoses of streptococcal angina, gastroenteritis, otitis media, and acute upper respiratory tract infections, i.e., those infectious diseases that are predominantly transmitted through direct contact through droplets, physical contact, or via the fecal–oral route. We suspect that the reduced direct contact between children due to the restricted attendance of educational facilities and the closure of recreational facilities has led to a greatly reduced transmission of infections. The proportion of children attending day care in Germany increased sharply from the 1–2 year age group to the 3–5 year age group [12], which is associated with a comparatively greater decrease in streptococcal angina and gastroenteritis. This suggests a substantial contribution to the transmission of these infections in day care settings. In contrast to gastroenteritis, the magnitude of the decline in streptococcal angina is the same for all age groups between 3 and 12 years. Infectious middle ear disease and upper respiratory tract infections particularly affect children up to and including 5 years of age (Table 1), who show a greater decline in these types of infections than older children, who are generally less affected by these infections. Consistent with our findings, an analysis of diagnoses in six pediatric emergency departments in France after the COVID-19 pandemic showed a sharp decline in those infections that are typically transmitted by direct contact, including those transmitted by droplet or fecal–oral routes, whereas the incidence of urinary tract infections remained unchanged [13].

While there was little change in the number of outpatient visits for injuries among children up to and including 5 years of age, they decreased by 20% among 6–10-year-olds and by as much as one-third among 11–12-year-olds (Table 1). We suspect that the restrictions placed on the independent activities of school children during the pandemic-related limitations reduced the risk of accidents and injuries.

Contrary to our expectations, there was no increase in outpatient visits associated with diagnosed mental and behavioral disorders in spite of potential increased stress on children caused by limited access to day care centers, schools, and leisure activities as well as peer to peer contacts. It is possible that for some children, higher stress was offset by more time spent at home and together with parents and other family, and this provided added interaction and attention. It must also be taken into account that mental disorders may not yet have clinically manifested during and shortly after the first lockdown and might only have led to a doctor’s diagnosis after a longer time interval. Other studies reported an increase in self-reported psychological stress and anxiety among children and adolescents [14], but no increase in diagnosed mental illnesses. In the U.S., emergency room visits for mental health disorders decreased by about 23% with the COVID-19 pandemic [15]. Among 11- to 16-year-olds in the United Kingdom, reported incidence of loneliness increased after COVID-related contact restrictions, but there was no observed increase in mental health disorders [16]. A survey of global data on the incidence of the COVID-19 pandemic on violence against children found conflicting results but no firm evidence of an increase [17]. In a study of more than 2500 adults in Germany, Sachser et al. even reported a decreased frequency of mental health problems during the first wave of the COVID-19 pandemic [18].

A strength of our study is that it is based on a very large database covering the medical visits of all children with statutory health insurance in Germany before and after the implementation of the first contact restrictions due to the COVID-19 pandemic. A potential weakness is the evaluation of the documented ICD codes, with possible residual imprecision of the codings. The data were collected during the first wave of COVID-19 infections in Germany and cannot be directly extrapolated to the effects of the subsequent second and third waves of infection and the contact restrictions implemented during those waves.

## 5. Conclusions

The impact of the COVID-19 pandemic and the related contact restriction measures did not markedly affect the outpatient care of children with chronic and serious health conditions. We found no detectable increase in outpatient visits associated with mental and behavioral disorders.

The sharp decline in children’s outpatient visits for infectious diseases points to an important contribution of infection transmission in day care centers, schools, and recreational facilities, which have been documented in previous studies, particularly for the day care sector [19,20]. We suggest that measures to prevent the transmission of infections should receive more attention after the pandemic. The risk of infection transmission in day care centers depends on their conditions, such as the space provided per child, the numerical ratio of children to caregivers and their qualifications, and the implemented hygiene measures [20,21]. More consistent training and empowerment of day care teachers and day care providers on health promotion measures for day care children, for example by online learning such as “Kinder gesund betreut” (https://kinder-gesund-betreut.de/; accessed on 10 July 2021), is desirable and should be further promoted. In Germany, federal legislation on day care described child health as an independent field of action for good day care quality [22], but so far, none of the 16 German states have chosen to apply for the available federal funds to improve this particular field. Smaller day care groups and larger outdoor areas are two approaches that have long been known to reduce the risk of infection [20,23,24]. Among children younger than 3 years of age in day care centers, an increased incidence of the six most common infectious diseases (upper respiratory tract infections, gastroenteritis, otitis media, ocular infections, tonsilitis, and bronchitis) has been shown to be related to confined spaces, the absence of mechanical ventilation, and the absence of an dedicated hygiene area [23]. After the pandemic, schools should also pay more attention to protecting and promoting child health, for example by regularly recording the immunization status of students and using appropriate hygiene approaches as well as effective air circulation and filtration systems. Regular consultation and support for schools by medically trained professionals is strongly encouraged in view of the reported large benefits: in Norway, a near doubling of the number of nurses with additional training in health promotion and disease prevention assigned to schools was associated with a reduced incidence of outpatient treatment, fewer adolescent pregnancies, and a significantly increased likelihood of successfully completing school and entering and completing university education [25].

The data presented here point to the great importance of direct contact between children for the transmission of infectious diseases. The potential for effective preventive measures should be utilized because reduced infections lead to an improved quality of life for the affected children, reduce the burden on their families and the loss of work time for caring parents, and they can contribute to educational success and thus to chances in life through the reduced absenteeism of children attending educational institutions.

## Figures and Tables

**Figure 1 children-08-00728-f001:**
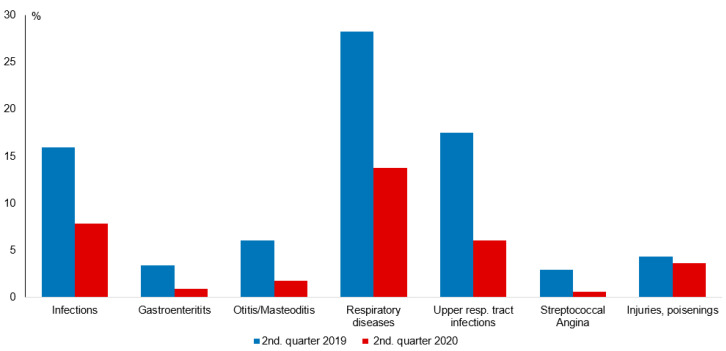
Frequency of diagnosed acute physical illnesses (% of all children) in children aged 1–12 years in the 2nd quarters of 2019 (control) and 2020 (COVID-19 pandemic).

**Figure 2 children-08-00728-f002:**
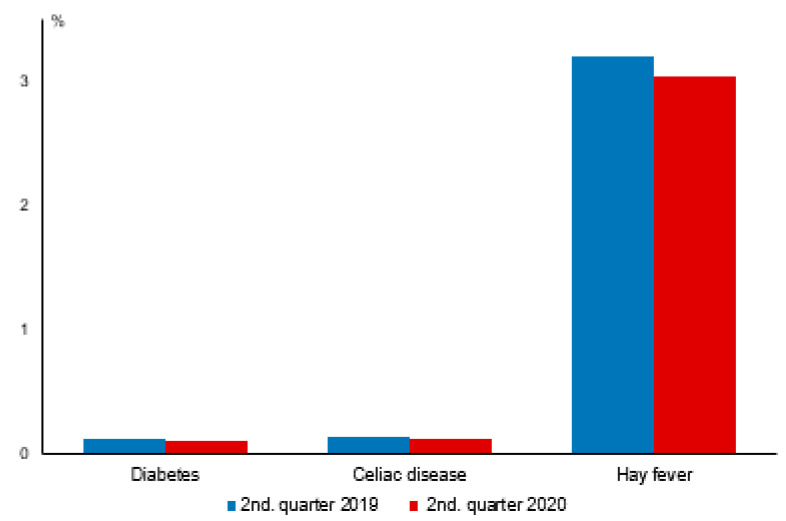
Frequency of diagnosed chronic physical conditions (% of all children) among children aged 1–12 years in the 2nd quarters of 2019 (control) and 2020 (COVID-19 pandemic).

**Figure 3 children-08-00728-f003:**
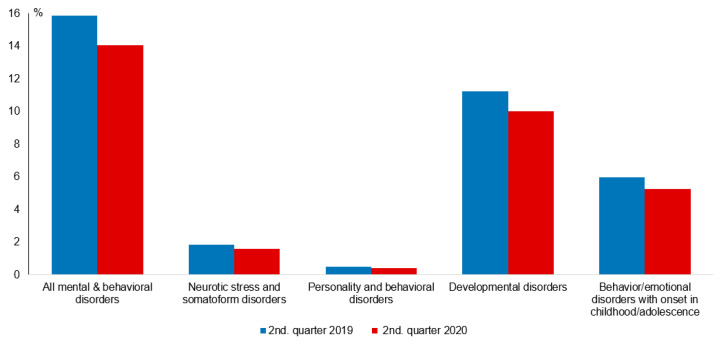
Frequency of diagnosed mental illnesses and behavioral disorders (% of all children) among children aged 1–12 years in the 2nd quarters of 2019 (control) and 2020 (COVID-19 pandemic).

**Table 1 children-08-00728-t001:** Number of outpatient doctor’s visits per child and frequency of selected diagnoses (ICD 10 codes) during Q2 2019 (control) and Q2 2020 (pandemic) for all children in Germany with statutory health insurance who presented to a physician’s office between January 2019 and June 2020, categorized by age group. SD = standard deviation, CI = confidence intervals, OLS = binary Ordinary Least Square model.

Age	1–2 Years			3–5 Years			6–10 Years			11–12 Years		
Year	2019	2020	P CI	2019	2020	P CI	2019	2020	P CI	2019	2020	P CI
	Raw means ± SD		OLS	Raw means ± SD		OLS	Raw means ± SD		OLS	Raw means ± SD		OLS
Children	1,529,063	1,617,556		2,194,429	2,241,386		3,278,228	3,351,685		1,289,184	1,288,880	
**Outpatient visits per child**	0.8446 ± 0.4417	0.7070 ± 0.4939	<0.0001 (−0.14, −0.13)	0.7430 ± 0.4867	0.5981 ± 0.5210	<0.0001 (−0.15, −0.14)	0.6654 ± 0.5077	0.5513 ± 0.5277	<0.0001 (−0.12, −0.11)	0.6377 ± 0.5147	0.5214 ± 0.5317	<0.0001 (−0.12, −0.11)
**Infections**												
Infectious and parasitic diseases	0.2297 ± 0.4206	0.1060 ± 0.3087	<0.0001 (−0.13, −0.12)	0.1730 ± 0.3783	0.0713 ± 0.2573	<0.0001 (−0.10, −0.10)	0.1369 ± 0.3437	0.0759 ± 0.2648	<0.0001 (−0.06, −0.06)	0.1096 ± 0.3123	0.0621 ± 0.2413	<0.0001 (−0.05, −0.05)
Infectious intestinal diseases	0.0512 ± 0.2205	0.0149 ± 0.1213	<0.0001 (−0.04, −0,04)	0.0383 ± 0.1920	0.0089 ± 0.0939	<0.0001 (−0.03, −0.03)	0.0274 ± 0.1632	0.0073 ± 0.0849	<0.0001 (−0.02, −0.02)	0.0235 ± 0.1515	0.0067 ± 0.0815	<0.0001 (−0.02, −0.02)
Diseases of the middle ear and mastoid	0.0773 ± 0.2670	0.0172 ± 0.1301	<0.0001 (−0.06, −0.06)	0.1002 ± 0.3003	0.0280 ± 0.1650	<0.0001 (−0.07, −0.07)	0.0416 ± 0.1997	0.0151 ± 0.1218	<0.0001 (−0.03, −0.03)	0.0207 ± 0.1423	0.0077 ± 0.0873	<0.0001 (−0.01, −0.01)
Diseases of the respiratory system	0.3730 ± 0.4836	0.1335 ± 0.3401	<0.0001 (−0.24, −0.23)	0.3230 ± 0.4676	0.1429 ± 0.3500	<0.0001 (−0.19, −0.18)	0.2393 ± 0.4267	0.1390 ± 0.3460	<0.0001 (−0.10, −0.10)	0.2174 ± 0.4125	0.1322 ± 0.3387	<0.0001 (−0.09, −0.08)
Acute upper respiratory tract infections	0.2699 ± 0.4439	0.0889 ± 0.2847	<0.0001 (−0.18, −0.18)	0.2084 ± 0.4061	0.0679 ± 0.2516	<0.0001 (−0.14, −0.14)	0.1344 ± 0.3411	0.0506 ± 0.2192	<0.0001 (0.08, −0.08)	0.1089 ± 0.3115	0.0365 ± 0.1874	<0.0001 (−0.07, −0.07)
Streptococcus Angina	0.0252 ± 0.1566	0.0069 ± 0.0825	<0.0001 (−0.02, −0.02)	0.0381 ± 0.1915	0.0082 ± 0.0902	<0.0001 (−0.03, −0.03)	0.0288 ± 0.1671	0.0061 ± 0.0780	<0.0001 (−0.02, −0.02)	0.0199 ± 0.1395	0.0044 ± 0.0659	<0.0001 (−0.02, −0.02)
**Injuries**												
Injuries	0.0402 ± 0.1965	0.0362 ± 0.1869	<0.0001 (−0.00, −0.00)	0.0387 ± 0.1929	0.0377 ± 0.1904	<0.0001 (−0.01, −0.01)	0.0434 ± 0.2039	0.0347 ± 0.1830	<0.0001 (−0.01, −0.01)	0.0564 ± 0.2306	0.0377 ± 0.1904	<0.0001 (−0.02, −0.02)
**Chronic physical** **diseases**												
Diabetes	0.0002 ± 0.0124	0.0001 ± 0.0114	n.s. (−0.00, 0.00)	0.0005 ± 0.0234	0.0005 ± 0.0232	n.s. (−0.00, −0.00)	0.0015 ± 0.0392	0.0015 ± 0.0381	0.0024 (−0.00, −0.00)	0.0028 ± 0.0532	0.0026 ± 0.0510	0.0005 (−0.00, −0.00)
Celiac disease	0.0005 ± 0.0221	0.0004 ± 0.0206	n.s. (−0.00, 0.00)	0.0011 ± 0.0331	0.0009 ± 0.0307	<0.0001 (−0.00, −0.00)	0.0018 ± 0.0428	0.0016 ± 0.0394	<0.0001 (−0.00, −0.00)	0.0022 ± 0.0465	0.0019 ± 0.0430	<0.0001 (−0.00, −0.00)
Hay fever	0.0033 ± 0.0572	0.0023 ± 0.0479	<0.0001 (−0.00, −0.00)	0.0138 ± 0.1167	0.0140 ± 0.1176	n.s. (−0.00, 0.00)	0.0444 ± 0.2061	0.0431 ± 0.2032	<0.0001 (−0.00, −0.00)	0.0662 ± 0.2486	0.0609 ± 0.2391	<0.0001 (−0.01, −0.01)
**Mental and** **Behavioral Disorders**												
All mental and behavioral disorders	0.0670 ± 0.2500	0.0628 ± 0.2426	<0.0001 (−0.01, −0.00)	0.1658 ± 0.3719	0.1513 ± 0.3583	<0.0001 (−0.02, −0.01)	0.1960 ± 0.3969	0.1717 ± 0.3771	<0.0001 (−0.03, −0.02)	0.1595 ± 0.3661	0.1389 ± 0.3458	<0.0001 (−0.02, −0.02)
Neurotic, stress and somatoform disorders	0.0061 ± 0.0780	0.0058 ± 0.0758	<0.0001 (−0.00, −0.00)	0.0109 ± 0.1038	0.0097 ± 0.0982	<0.0001 (−0.00, −0.00)	0.0235 ± 0.1515	0.0205 ± 0.1418	<0.0001 (−0.00, −0.00)	0.0336 ± 0.1803	0.0282 ± 0.1656	<0.0001 (−0.01, −0.01)
Personality and behavioral disorders	0.0012 ± 0.0344	0.0011 ± 0.0333	n.s. (−0.00, −0.00)	0.0042 ± 0.0644	0.0036 ± 0.0598	<0.0001 (−0.00, −0.00)	0.0066 ± 0.0812	0.0053 ± 0.0729	<0.0001 (−0.00, −0.00)	0.0069 ± 0.0826	0.0055 ± 0.0741	<0.0001 (−0.00, −0.00)
Developmental Disabilities	0.0481 ± 0.2140	0.0449 ± 0.2072	<0.0001 (−0.00, −0.00)	0.1381 ± 0.3450	0.1264 ± 0.3323	<0.0001 (−0.01, −0.01)	0.1366 ± 0.3434	0.1197 ± 0.3246	<0.0001 (−0.02, −0.02)	0.0820 ± 0.2744	0.0727 ± 0.2596	<0.0001 (−0.01, −0.01)
Behavioral and emotional disorders with onset in childhood & adolescence.	0.0124 ± 0.1107	0.0119 ± 0.1085	<0.0001 (−0.00, −0.00)	0.0353 ± 0.1846	0.0323 ± 0.1768	<0.0001 (−0.00, −0.00)	0.0842 ± 0.2777	0.0744 ± 0.2624	<0.0001 (−0.01, −0.01)	0.0936 ± 0.2913	0.0832 ± 0.2762	<0.0001 (−0.01, −0.01)

## Data Availability

The data used are owned and controlled by *Kassenärztliche Bundesvereinigung*, Herbert-Lewin-Platz 2, 10623 Berlin, Germany.

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
