# Peer review of "COVID-19 Associated Contact Restrictions in Germany: Marked Decline in Children’s Outpatient Visits for Infectious Diseases without Increasing Visits for Mental Health Disorders"

_children, 2021, doi:10.3390/children8090728_

Round 1

Reviewer 1 Report

Thank you for the opportunity to review this paper. I am sorry to say that I found it very difficult to understand, and would need the paper to be significantly revised for me to provide a sufficient review. As starting points:

  • The analysis section could be developed more. E.g. how does OLS take into account control or pandemic groups - what does this mean? Who are the people missed in the dataset and how do they differ from those included? Is the group missed different between the two years and, if so, should their characteristics be controlled for in the analyses?
  • The graphs don't appear to reflect the Table or text content. The figures appear to be replicated. The y axis doesn't appear to reflect the %. 
  • I think it would help if the differences between quarters and their confidence intervals are displayed, and these are referred to in text, with reference to % of control.  
  • Interpretation: If some medical diagnoses are decreasing between the years but mental health diagnoses are not - does this mean that in relative terms the mental health problems could be increasing? 

Author Response

Thank you for the opportunity to review this paper. I am sorry to say that I found it very difficult to understand, and would need the paper to be significantly revised for me to provide a sufficient review.

Thank you very much indeed for your constructive comments and for giving us the opportunity to resubmit a revised manuscript. We have tried to present the information more clearly and thoroughly addressed all issues raised, and we very carefully dealt with those issues that warranted special attention.

The analysis section could be developed more. E.g. how does OLS take into account control or pandemic groups - what does this mean? Who are the people missed in the dataset and how do they differ from those included? Is the group missed different between the two years and, if so, should their characteristics be controlled for in the analyses?

Thank you for pointing this out. The following answers to your suggestions apply.

    • In the OLS regression, a dummy variable that indicates whether the observation was in 2020 (pandemic group) or in 2019 (control group) was regressed on the respective health outcome variables. In addition, age, gender, month of birth and county fixed effects were included as control variables.
    • The dataset captures all children within the statutory health insurance system who at least once visited a medical doctor’s office between January 2019 and June 2020. The share of children with statutory health insurance is fairly constant over time, for 2019 and 2020 it amounts to about 90% of children between 0 and 15 years. This number is calculated by comparing official birth record numbers from children born in the relevant birth cohorts (2005-2019 for 2019 and 2006-2020 for 2020) with the number of children insured via the statutory health insurance system in the corresponding years. Thus, our analysis only misses children that are either privately insured or who are in the statutory health insurance system but did not visit a doctor’ office between January 2019 and June 2020. Particular for younger children this are rare cases, as the German health care system requires regular health checks for children. Independent of this, the groups missed are the same for both years respectively periods. Thus, from a methodological point of view it does not matter if the populations per se matter in respect to their socio-economic background as long as the changes between the years do not differ systematically. We have no reason to assume this.
    •  
  • The graphs don't appear to reflect the Table or text content. The figures appear to be replicated. The y axis doesn't appear to reflect the %. 
    • Thank you for pointing this out. Indeed a mistake occured when insertign the figures, by which Figure 1 erroneously was the same as Figure 3.  We corrected this and have redrawn all figures to add clarity.
    • The y-axis does indeed display the prevalence of the diseases in %. which is now indicated in the figures For example, in Figure 3, about 7% of children had a mental and behavioral disorders diagnosis both in 2019 and 2020. We clarified this also in the text.

I think it would help if the differences between quarters and their confidence intervals are displayed, and these are referred to in text, with reference to % of control.  

Thanks for this helpful comment.

    • We added 95% Confidence Intervals obtained in the OLS Regressions to Table 1. Instead of adding confidence intervals in the figures, we constructed a new Table (Table A.1 in the Appendix) including means, SD, p-values and CIs for the results for 1-12 year old children. However, we did not include an interpretation of the confidence intervals in the text, as they are very small which emphasizes the accuracy of our estimations
    • The %-numbers reported in the text are relative to the control group. For example, treatment cases fell from 0.71 in 2019 to 0.59 in 2020. 0.59 is 82% of 0.71.

Interpretation: If some medical diagnoses are decreasing between the years but mental health diagnoses are not - does this mean that in relative terms the mental health problems could be increasing? 

Thanks for raising this question. In the data, each diagnosis children got during the observation period is recorded. The displayed averages represent the share of children that have a relevant diagnosis during the observation period. Thus, the average shares of children with different diagnoses are independent from each other. We tried to clarify this in the text.

Reviewer 2 Report

This is a highly interesting and innovative paper presenting a hugh amount of data and prospectively evaluating mental and inflammatory disorders in the context of the Covid-19 pandemic. It certainly shows a paradigm change which is underlined by data in ca. 8 Mio children that the outcome in children during the pandemic was more positive than expected concerning mental, infectiological and social situation. These results have a strong positive impact on poossible targeted education and counseling in the future and shopw that extensive relationships can also be harmful, however attenuated measured might be evaluated in the future protecting the health of children by reduced social stress.

Author Response

This is a highly interesting and innovative paper presenting a hugh amount of data and prospectively evaluating mental and inflammatory disorders in the context of the Covid-19 pandemic. It certainly shows a paradigm change which is underlined by data in ca. 8 Mio children that the outcome in children during the pandemic was more positive than expected concerning mental, infectiological and social situation. These results have a strong positive impact on posssible targeted education and counseling in the future and show that extensive relationships can also be harmful, however attenuated measured might be evaluated in the future protecting the health of children by reduced social stress.

Thank you for the kind and encouraging comment that is greatly appreciated. We have thoroughly revised the manuscript and believe it has been further improved.